# Preparation of Ganoderma Lucidum Bran-Based Biological Activated Carbon for Dual-Functional Adsorption and Detection of Copper Ions

**DOI:** 10.3390/ma16020689

**Published:** 2023-01-10

**Authors:** Baoying Wang, Jingming Lan, Chunmiao Bo, Bolin Gong, Junjie Ou

**Affiliations:** 1School of Chemistry and Chemical Engineering, Key Laboratory for Chemical Engineering and Technology, State Ethnic Affairs Commission, Ningxia Key Laboratory of Solar Chemical Conversion Technology, North Minzu University, Yinchuan 750021, China; 2CAS Key Laboratory of Separation Science for Analytical Chemistry, Dalian Institute of Chemical Physics, Chinese Academy of Sciences, Dalian 116023, China

**Keywords:** activation, biomass AC, carbon dots, detection, dua-functional material

## Abstract

In this paper, Ganoderma lucidum bran was explored as the precursor to fabricate biomass activated carbon. When potassium hydroxide was selected as an activator (1:6, mass ratio of AC-12 to potassium hydroxide), and the activation condition was 700 °C at 5 h, the highest specific surface area reached 3147 m^2^ g^−1^. Carbon dots were prepared with citric acid monohydrate and thiourea as precursors and then loaded onto the surface of activated carbon by a simple and green method. Activated carbon for dual-functional had a high adsorption capacity. Additionally, based on its unique optical properties, the fluorescence response for detecting copper ion was established. The fluorescence intensity of the materials decreased linearly with the increase of copper ion concentration, in the range of 10–50 nmol L^−1^. The research opened up a new way for applying biomass activated carbon in the field of adsorption and detection. Highlights: (1) Carbon dots were loaded on the surface of activated carbon; (2) the simultaneous adsorption and detection were realized; (3) it provides a way for the preparation of dual-functional materials.

## 1. Introduction

Mushroom bran, also known as mushroom residue, is the residue of the culture medium after harvesting mushrooms, which is made of raw materials such as straw and sawdust. After the edible fungi harvest, many mycelium and beneficial bacteria are left on the stick, and a variety of sugars, organic acids, enzymes, and bioactive substances are produced through enzymatic hydrolysis during the growth of mycelium. The bacterial residue is rich in protein, cellulose, and amino acids [1]. For example, Ganoderma lucidum bran (GB) residue contains not only a large number of mycelium and protein but also trace elements such as iron, calcium, zinc, and magnesium [2]. Improper disposal will cause a waste of resources and seriously destroy the environment for human survival and health.

Activated carbon (AC) has a hierarchical structure, high surface area and porosity, and a variety of surface-active sites, and it is widely employed as an adsorbent to effectively remove pollution. However, the regeneration process of industrial AC is expensive and complex, which makes the application of this material economically infeasible. To address these issues, a plethora of precursor materials have been selected to prepare AC. For example, Almahbashi et al. selected sewage sludge as raw material to produce AC with a maximum surface area of 377.7 m^2^ g^−1^ [3]. In addition, Sun et al. prepared AC with coffee shells as a precursor, and the surface area was 2349 m^2^ g^−1^ [4]. These works offer us a hint to utilize the agricultural waste from GB for the preparation of AC. On the one hand, the rational utilization of GB can reduce environmental pressure and create economic value. On the other hand, developing a new method of preparing AC by GB can reduce costs [5]. However, the single functional carbon-based adsorbent can only achieve the function of turning waste into treasure and has certain limitations for treating heavy metals [6]. Therefore, preparing a bifunctional carbon-based material that can adsorb and detect heavy metals is necessary.

Carbon dots (CDs) are composed of ultrafine, quasi-spherical carbon nanoparticles with a size of less than 10 nm [7]. CDs have good water solubility, low toxicity, and excellent optical properties, and have been widely employed in many fields [8]. Due to these characteristics, preparing dual-functional materials has become an essential target for adsorption and detection. However, very little has been reported on such dual-functional materials. To realize the simultaneous detection and adsorption performance of GB biomass AC, the detection of metal ions is based on the fluorescence quenching phenomenon of CDs combined with metal ions, and the adsorption of heavy metals can be realized via the interaction between the active site of AC and heavy metal ions. Herein, the AC with a large specific surface area was prepared from GB and then integrated with CDs via a simple impregnation method. The detection and their adsorption capacity of heavy metals were studied.

## 2. Experimental Section

### 2.1. Reagents and Materials

Potassium hydroxide (KOH), potassium permanganate (KMnO_4_), zinc chloride (ZnCl_2_), citric acid monohydrate (C_6_H_10_O_8_), and thiourea (CH_4_N_2_S) were provided by Aladdin Industries Co. (Shanghai, China). ZnCl_2_, CaCl_2_, LiCl, Cd(NO_3_)_2_·4H_2_O, CoCl_2_·6H_2_O, Cr(NO_3_)_3_·9H_2_O, CuSO_4_, FeCl_3_·6H_2_O, KCl, Mg(NO_3_)_2_, NaCl, Pb(NO_3_)_2_, and ZrCl_4_ were the prepared solutions of metal ions (Zn^2+^, Ca^2+^, Li^+^, Cd^2+^, Co^2+^, Cr^3+^, Cu^2+^, Fe^3+^, K^+^, Mg^2+^, Na^+^, Pb^2+^ and Zr^4+^), which were also from Aladdin Industries Co. (Shanghai, China). Nitric acid (HNO_3_), hydrochloric acid (HCl), and absolute ethanol (C_2_H_6_O_2_) were provided by Sinopharm Chemical Reagent Co., Ltd. (Shanghai, China). GB was obtained from Ningxia Academy of Agricultural and Forestry Sciences (Yinchuan, China).

### 2.2. Preparation Process

#### 2.2.1. Preparation of AC

The GB was crushed and passed through a 30 mesh screen to obtain a powder. It was placed in a tubular furnace and carbonized at 500 °C for 3 h under the N_2_, and the AC precursor was obtained.

In order to optimize the activation conditions of the AC, single-factor experiments were conducted to explore the effects of the proportion of activators, type of activator, activation temperature, and time on a specific surface area.

#### 2.2.2. Preparation of AC@CD

The CDs were prepared according to the previous report [9]. In brief, 1.26 g of C_6_H_10_O_8_ and 1.37 g of CH_4_N_2_S were dissolved in water and put in a reaction vessel for 4 h at 160 °C. Then, 0.01 g of AC reacted with 0.01 g of CDs (after freeze-drying for 24 h) at 25 °C for 24 h. The resulting product was dried to a constant weight for 24 h at 60 °C and named as AC@CD. The synthetic schematic of AC@CD is shown in Figure 1.

### 2.3. Characterization of Material

The nitrogen adsorption-desorption isotherm and pore size distribution of the adsorbent were measured by Brunner-Emmet-Teller (BET, Tri star 3020, McMurray Tec Instrument Co., Ltd., Canonsburg, PA, USA). The surface functional groups of the adsorbent were determined by Fourier-transform infrared spectroscopy (FT-IR, Thermo Nicolet iS50 spectrometer, Thermo Fisher Scientific Co., Ltd., Waltham, MA, USA). X-ray diffraction was obtained by scanning with Smartlab SE (XRD, XRD-6100200 Ma, Shimadzu Enterprise Management Co., Ltd., Kyoto, Japan). The morpho-ogy of the adsorbent was determined by scanning electron microscopy (SEM, JEM-7500F, Japan Electronics Co., Ltd., Tokyo, Japan). A nanoparticle size and zeta potential analyzer (Malvern Zetasizer nano ZS90, Malvern Instruments Co., Ltd., Malvern, Worcestershire, UK) was employed to measure the potential of the sample. The steady state and transient fluorescence spectrometer (RF, Fluoro Max-4, HORIBA Co., Ltd., Irvine, CA, USA) measured its fluorescence emission spectrum. The residual concentration was analyzed using an ultraviolet-visible spectrophotometer (UV, TU-1950, Beijing General Instrument Co., Ltd., Beijing, China).

## 3. Results and Discussion

### 3.1. Preparation and Characterization of AC@CD

The AC precursor was prepared with GB as a biomass source by simple carbonization. The chemical activation method mixes the activator and carbonized biomass in a particular proportion, then reacts at a high temperature. Activators can be classified as alkaline, acidic, neutral, and so on. Different activators, ratio of precursor and activator, and activation, activation temperature, and time will remarkably affect both physical and chemical properties of as-synthesized AC [10,11]. Herein, the AC precursor with GB as the raw material was prepared, and the optimal activation conditions were explored. The CDs with good fluorescence performance reacted with the obtained product to obtain a dual-functional material.

A series of ACs were obtained under different conditions, as listed in Table 1. KOH was selected as an activator, and the activation temperature was set at 300 °C, and the activation time at 3 h under N_2_. The effect of the mass ratio of the AC precursor to the KOH (1:2, 1:4, 1:6, and 1:8) was investigated, and the obtained ACs were assigned as AC-1, AC-2, AC-3, and AC-4. The specific surface area of the AC increased from 12 to 1633 m^2^ g^−1^, then decreased to 8.26 m^2^ g^−1^. Different amounts of activator can change the physical properties of AC, allowing the specific surface area to increase as the activator increases. However, too many activators can lead to the collapse of pore space, decreasing the particular surface area.

The effect of activator type such as KMnO_4_, ZnCl_2,_ and HNO_3_ on the specific surface area and pore structure of AC were also investigated, and the activation was performed under N_2_ at 300 °C for 3 h (1:6, mass ratio of AC precursor to KOH). The obtained ACs were named as AC-5, AC-6, and AC-7. The specific surface area of the AC fabricated with ZnCl_2_ was 18.0 m^2^ g^−1^, and that of the AC with KMnO_4_ was 832 m^2^ g^−1^. The specific surface area of the AC fabricated with HNO_3_ was 21.55 m^2^ g^−1^, and that of the AC with KOH was 1633 m^2^ g^−1^, which was the highest surface area using four kinds of activators. Potassium hydroxide could inhibit the formation of tar, thus achieving a good activation effect at a low reaction temperature.

The influence of activation temperature (500, 700, and 900 °C) on AC was investigated by keeping the mass ratio of the AC to KOH at 1:6 under N_2_, with activation for 3 h. The obtained samples were named AC-8, AC-9, and AC-10. The specific surface area of the carbon first increased and then decreased with an increase in activation temperature, acquiring a maximum specific surface area of 2155 m^2^ g^−1^ at 700 °C. The effect of activation time (1, 5, and 7 h) was also studied by maintaining the activation temperature at 700 °C, and the resulting ACs were named AC-11, AC-12, and AC-13. The specific surface area of the carbon first increased and then decreased, and the maximum specific surface area reached 3147 m^2^ g^−1^ (AC-12).

It can be observed from Figure 2a that there was an apparent H4 hysteresis loop in the region of P/P_0_ = 0.3–1.1, representing capillary condensation in the mesopore. At the low-pressure profile (P/P_0_ = 0–0.3), a gentle inflection point formed by single-layer dispersion and the central location with a slight slope were obtained by multi-layer diffusion (Figure 2b). So, the adsorption and desorption isothermal curve belongs to the type IV isothermal curve. The pore size of the different ACs was 2–21 nm, indicating the existence of mesopores in the prepared AC.

In our case, four kinds of GB-based ACs were modified with CDs. As shown in Table 1, the specific surface area of AC-1@CD changed to 1510 m^2^ g^−1^ after modification, and that of AC-4@CD increased to 135.2 m^2^ g^−1^. The specific surface area of AC-9@CD changed to 2580 m^2^ g^−1^, and that of AC-12@CD changed to 3284 m^2^ g^−1^. Obviously, their specific surface area slightly increased after modification with CDs, proving that the CDs were successfully loaded onto the AC [11,12].

Figure 3 shows SEM images of AC-12 and AC-12@CD. Honeycomb-like porous structures existed, indicating that GB was a suitable carbon source for preparing porous materials [13]. Additionally, there were no significant differences between AC-12 and AC-12@CD due to the large size of the CDs, which could not enter the micropores in the AC [14].

FT-IR spectra of AC-12, AC-12@CD, and AC-12@CD-Cu^2+^ (AC-12@CD after adsorption of Cu^2+^) are shown in Figure 4a. Wide peaks at 3428 cm^−1^ were assigned as a tensile vibration of -NH_x_ or -OH, while the peak at 1623 cm^−1^ was assigned as a bending vibration of C=C [15]. Their intensities in AC-12@CD was significantly enhanced relative to those in AC-12, indicating the successful loading of CDs onto the AC-12 surface. The absorption peaks at 1017 and 1073 cm^−1^ were asymmetric stretching vibration peaks of C–O and C–H [16,17]. The peaks at 3428 and 1073 of AC-12@CD-Cu^2+^ were obviously weakened, because the materials had a complexation reaction with Cu^2+^. All this evidence indicated the presence of a few oxygen-containing functional groups [18]. Many groups, such as amino, hydroxyl, and carboxyl groups in both AC-12 and AC-12@CD, would provide the basis to remove heavy metals and be suitable for application in water treatment.

The phase structure analysis of AC-12 and AC-12@CD was carried out using the XRD technique, as shown in Figure 4b. Two hump-shaped diffraction peaks were located at 22.24° and 43.12°. The former was assigned as characteristic peaks of the (002) and (100) crystal planes of the carbon material, indicating that the diffraction peaks of AC-12 and AC-12@CD were still mainly graphite-like phase g-C_3_N_4_ structures [19,20]. However, the peak intensities at 22.61° and 43.12° for AC-12@CD were reduced compared to AC-12, indicating an increased graying of AC-12@CD [21].

The surface property of AC-12 and AC-12@CD was also tested by employing a zeta potential meter at pH = 7. The zeta potential of AC-12 particles in water was −12.9 mV, but after loading the CDs, the zeta potential on the carbon surface became −21.3 mV. It was related to the increase of –NH or –NH_2_ and –OH functional groups on the surface of AC-12@CD, causing the surface to be more negatively charged [22]. This result also proved the successful loading of the CDs.

### 3.2. Adsorption Ability of AC@CD for Cu^2+^

The adsorption capacities of four kinds of AC@CD were measured under the same conditions. The adsorption capacity of AC-12@CD was 36.31 mg g^−1^, those of AC-1@CD, AC-4@CD, and AC-9@CD were 27.69, 13.27, and 30.58 mg g^−1^, respectively. The difference in the adsorption capacity was related to its specific surface area. The larger the specific surface area of carbon material, the richer the pore structure, and the higher the adsorption capacity of Cu^2+^ [23,24]. To sum up, the adsorption performance of AC-12@CD was the best, which was subsequently measured as an example.

#### 3.2.1. Isothermal Adsorption and Kinetic Experiment

The isothermal adsorption model of AC-12 and AC-12@CD were studied, as shown in Figure 5a. AC-12 and AC-12@CD were added into 200 mL of Cu^2+^ solutions with different initial concentrations. The adsorption amount of AC-12 increased at the initial Cu^2+^ concentrations of 90–160 μmol L^−1^, the adsorption reached the maximum at 170 μmol L^−1^, and that of AC-12 gradually equilibrated at 180–190 μmol L^−1^. The adsorption amount of AC-12@CD increased at the initial Cu^2+^ concentrations of 90–150 μmol L^−1^, the adsorption amount reached the maximum at 160 μmol L^−1^, and that of AC-12 gradually equilibrated at 170–190 μmol L^−1^. The kinetic adsorption model of AC-12 and AC-12@CD is shown in Figure 5b. The adsorption amount of AC-12 increased within 35 min, and the equilibrium time was determined to be 35 min. Additionally, that of AC-12@CD increased within 30 min, so the equilibrium time was 30 min. The adsorption time was shortened due to the increased adsorption sites of the modified materials. In the adsorption at 35 and 30 min, the adsorption was faster because the material was mainly a chemical reaction (complexation reaction). With the increase of time, the adsorption rate gradually slowed down because the adsorption site had reached saturation. A few materials are listed in Table 2. Runtime et al. [25] selected biomass carbon residue as a low-cost adsorbent to treat Cu^2+^ in water during biomass gasification, the adsorption capacity of which reached 23.10 mg g^−1^ at 120 min. The AC from waste rubber tires was prepared to adsorb Cu^2+^ in a polymetallic aqueous solution by Cherono et al. [26], and the adsorption capacity reached 12.44 mg g^−1^ at 120 min. The maximum adsorption capacity of AC prepared from Malawian monkey bread shells was 3.083 mg g^−1^ [27]. The adsorption capacity of AC prepared from pistachio fruit (277.8 mg g^−1^) was higher than that reported by us, but the adsorption equilibrium time was too long, requiring 180 min [28]. The adsorption capacity of AC-12 and AC-12@CD could reach 49.35 and 36.55 mg g^−1^. AC-12 and AC-12@CD showed the ability to rapidly adsorb Cu^2+^ in a solution and with a high adsorption capacity.

#### 3.2.2. Fitting of Isothermal Adsorption Experimental Data

The fitted curves of the Freundlich and Langmuir models for AC-12 and AC-12@CD are shown in Figure 5c,d, and the correlation coefficients are listed in Table 3. The correlation coefficients (R) of the Langmuir model were 0.9758 and 0.9791, respectively, which were remarkably higher than those of the Freundlich model (R = 0.9352, 0.9656). It confirmed that the theoretical model agrees better with the experimental data; the Langmuir model could be better described in the adsorption process than the Freundlich model. The adsorption characteristics of Cu^2+^ on AC-12 and AC-12@CD were of unimolecular adsorption with uniform distribution of active groups on the adsorbent surface [24,29]. The functional groups such as –NH_x_, –OH, and –COOH in the adsorbent could remove metal pollution through electronic interaction with charged metal ions in the solution [30,31].

#### 3.2.3. Dynamic Experiment Data Fitting

The quasi-first-order kinetic adsorption curve model for AC-12 and AC-12@CD are exhibited in Figure 5e,f, and the resulting model parameters are listed in Table 4. The quasi-second-order kinetic adsorption curve model (R = 0.9998, 0.9961) was more suitable for AC-12 and AC-12@CD. The calculated and actual values of the equilibrium sorption capacity were in good agreement and the proposed quasi-second-order kinetic curve model fits the experimental data well [32]. The adsorption was related to the chemisorption of metal ions by the reactive groups on the surface of the material [33,34].

### 3.3. Fluorescence Property of AC@CD

Although the specific surface areas of AC-1@CD, AC-4@CD, AC-9@CD, and AC-12@CD all increased, the fluorescence responses of them varied greatly. AC-1@CD hardly produced a fluorescence response, while AC-4@CD exhibited the highest fluorescence response. The pore size of AC-4 was too large to load some aggregate CDs, while AC-12@CD possessed the largest surface area, in which the micropores could not load the CDs with better dispersion. There was not enough pore capacity of AC-1@CD to load some aggregate CDs, and there were not enough micropores of AC-9@CD to load the well-dispersed CDs. This was the main reason for the poor or even no fluorescence response of AC-1@CD and AC-9@CD.

#### 3.3.1. Fluorescence Yield of AC@CD

Fluorescence yield is an essential parameter for evaluating the fluorescence ability of materials. Anthracene with a similar excitation wavelength of 366 nm was selected as the standard sample to calculate the fluorescence yield of AC@CD. The fluorescence yield of AC-4@CD and AC-12@CD were 2.63% and 0.42%, respectively.

#### 3.3.2. Fluorescence Selective and Sensitive Detection for Metal Ions

The selectivity of AC-12@CD for metal ions was investigated, as shown in Figure 6a. The fluorescence signals of Mg^2+^, Ca^2+^, Co^2+^, Zr^4+^, Cd^2+^, Zn^2+^, K^+^, Na^+^, Pd^2+^, and Cr^2+^ on AC-12@CD were negligible. The sensitivity of the fluorescence signal of AC-12@CD with Fe^3+^ and Li^+^ were 30% and 28%, also much lower than that of Cu^2+^ (46%). The strong response to Cu^2+^ was mainly attributed to the nitrogen-containing groups on the surface of AC-12@CD. Electrons were attributed to Cu^2+^ as electron donors, and the lone pair electrons of the nitrogen atom showed a good affinity for Cu^2+^ [35,36]. The quenching constant was an important means of determining fluorescence sensitivity, which were calculated according to the Stern-Volmer equation. The quenching constant of AC-4@CD and AC-12@CD was 5.63 × 10^6^ and 1.32 × 10^7^ L mol^−1^, respectively, demonstrating a perfect fluorescence response of AC@CD to the Cu^2+^ ion.

#### 3.3.3. Quantitative Measurement of AC@CD

The fluorescence responses of AC-4@CD and AC-12@CD to different concentrations of Cu^2+^ were tested. The fluorescence intensities of AC-4@CD and AC-12@CD decreased with an increase of Cu^2+^ concentration (Figure 6b,d). The linear relationship between the fluorescence intensity of AC-4@CD and AC-12@CD with Cu^2+^ concentration is shown in Figure 6c,e, and R was 0.9987 and 0.9887, respectively. The LOD of AC-4@CD and AC-12@CD were 11.40 and 16.95 nmol L^−1^, when the concentration of Cu^2+^ was in the range of 10–50 nmol L^−1^, which provided a new tool for the application of carbon-based dual-functional materials. Table 5 lists several CD materials with fluorescent properties for the selective detection of Cu^2+^. Ma et al. [37] prepared CDs using natural peanut shells as precursor materials; the LOD was 4.80 μmol L^−1^. Tan et al. [38] selected pyrolyzed sago waste to generate CDs as potential probes for metal ion sensing, with an LOD of 7.80 μmol L^−1^ in the concentration range of 0–48 μmol L^−1^. Zheng et al. [39] functionalized CDs and achieved an LOD of 5.00 μmol L^−1^ in the range of 0.5 to 10 μmol L^−1^. Hydrophobic CDs were also synthesized by the one-pot method, with an LOD of 0.2 μmol L^−1^ [40]. The LOD of AC@CD in detecting Cu^2+^ was much lower than other reported materials.

## 4. Conclusions

A facile and cost-effective strategy for dual-functional GB biomass AC was developed. The CDs with excellent optical property were loaded into the AC to generate the blue emission at 451 nm. The fluorescence emission of AC@CD could be selectively quenched by Cu^2+^. The AC@CD fluorescence detection method was constructed to selectively detect Cu^2+^ with a linear range from 10 to 50 nmol L^−1^. At the same time, the AC@CD possessed a high adsorption capacity. Both adsorption isotherms and kinetics curves confirmed that the adsorption of Cu^2+^ by AC@CD was monolayer adsorption via chemical interaction. As a result, the dual-functional GB biomass AC with the adsorption and detection of Cu^2+^ was successfully developed, which not only reasonably avoided the disposal of carbon dots, but also provided more possibilities for the application of carbon-base materials.

## Figures and Tables

**Figure 1 materials-16-00689-f001:**
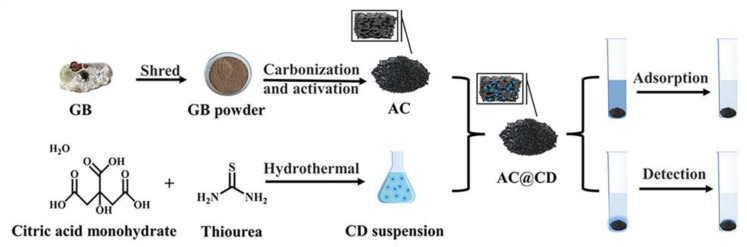
Synthetic schematic of AC@CD.

**Figure 2 materials-16-00689-f002:**
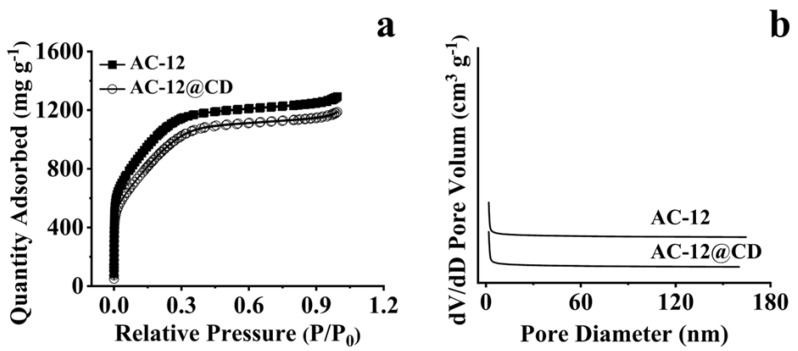
N_2_ adsorption analysis diagram of AC-12 and AC-12@CD. (**a**) Adsorption isotherm, (**b**) pore size distribution diagram.

**Figure 3 materials-16-00689-f003:**
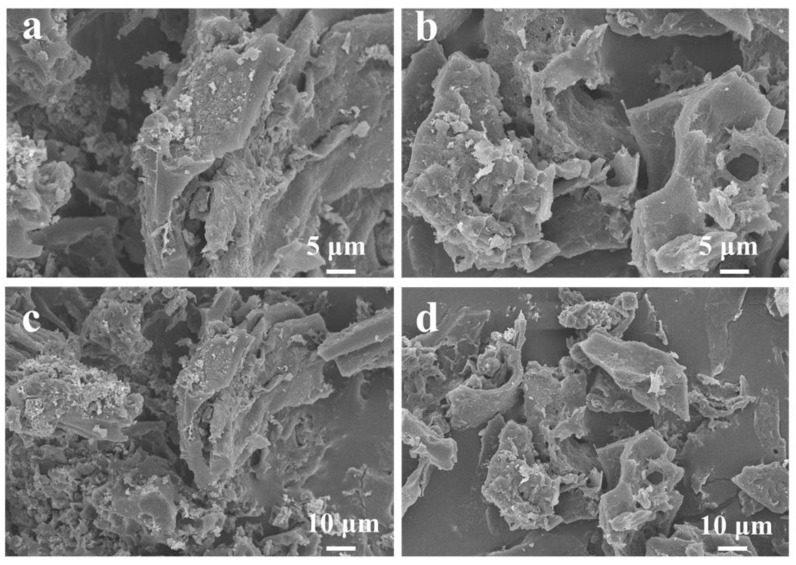
SEM images of (**a**,**c**) AC-12 and (**b**,**d**) AC-12@CD.

**Figure 4 materials-16-00689-f004:**
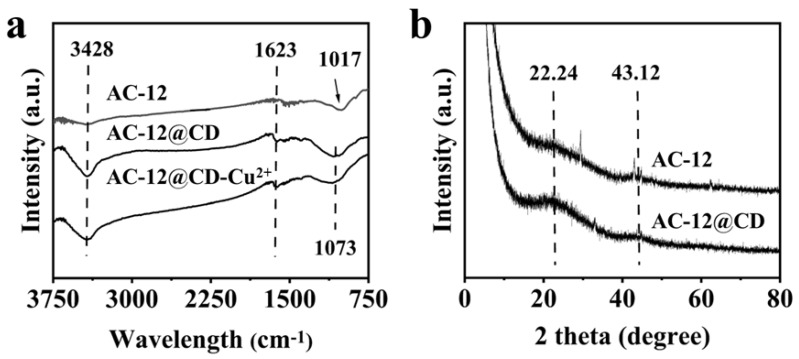
(**a**) FT−IR spectra of AC-12 and AC-12@CD, (**b**) XRD diffraction of AC-12 and AC-12@CD.

**Figure 5 materials-16-00689-f005:**
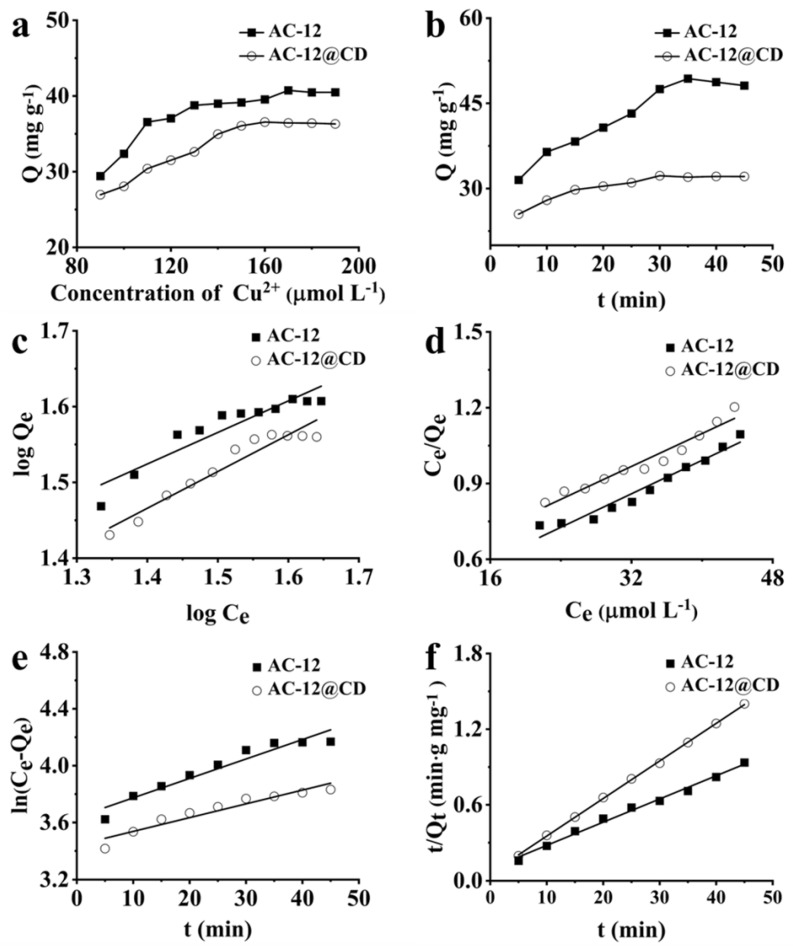
(**a**) Isothermal adsorption curve, (**b**) kinetic adsorption curve, (**c**) Freundlich curve, (**d**) Langmuir curve, (**e**) quasi-first-order kinetic curve, (**f**) quasi-second-order kinetic curve.

**Figure 6 materials-16-00689-f006:**
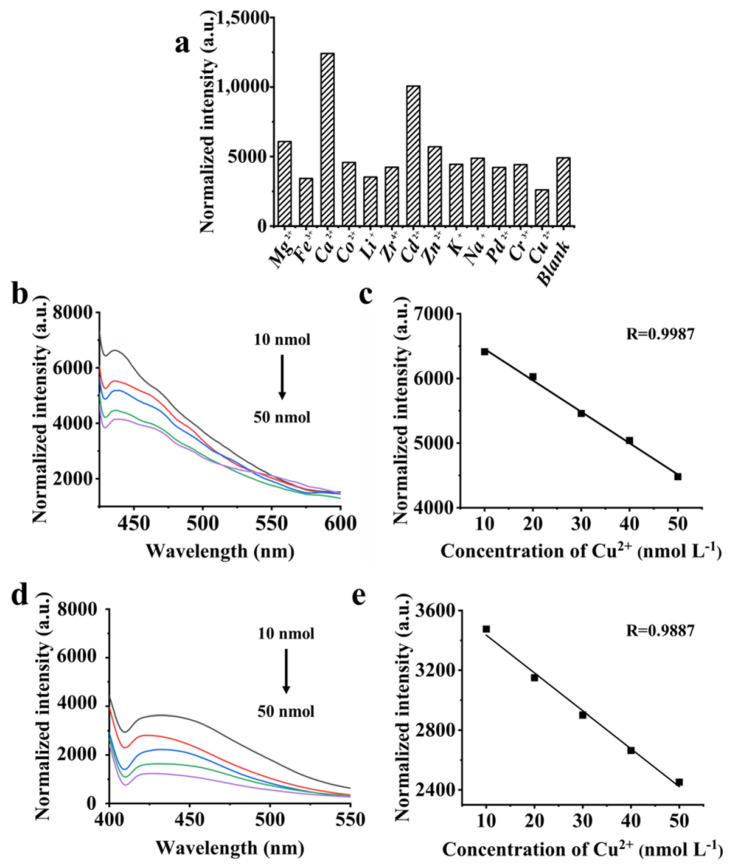
(**a**) Fluorescence intensity of AC-12@CD with 13 kinds of metal ions, (**b**) fluorescence emission spectrum of AC-4@CD containing different concentrations of Cu^2+^, (**c**) linear response curve of AC-4@CD for detection of Cu^2+^, (**d**) fluorescence emission spectrum of AC-12@CD containing different concentrations of Cu^2+^, (**e**) linear response curve of AC-12@CD for detection of Cu^2+^.

**Table 1 materials-16-00689-t001:** Surface area and porosity property of AC fabricated with different conditions.

Material	Activator	Mass Ratio	Temperature (°C)	Time (h)	S_BET_ (m^2^ g^−1^)	V_g_ (cm^3^ g^−1^)	D_pore_ (nm)
AC-precursor	-	-	300	3	12.0	0.02	12.6
AC-1	KOH	1:2	300	3	1148	0.17	3.94
AC-2	KOH	1:4	300	3	1498	0.78	2.42
AC-3	KOH	1:6	300	3	1633	0.81	2.24
AC-4	KOH	1:8	300	3	8.3	0.06	3.82
AC-5	KMnO_4_	1:6	300	3	832.0	0.11	9.04
AC-6	ZnCl_2_	1:6	300	3	18.0	0.03	3.65
AC-7	HNO_3_	1:6	300	3	21.55	0.02	5.01
AC-8	KOH	1:6	500	3	1441	0.34	2.24
AC-9	KOH	1:6	700	3	2155	0.64	2.74
AC-10	KOH	1:6	900	3	875.0	0.74	3.44
AC-11	KOH	1:6	700	1	2779	0.68	2.78
AC-12	KOH	1:6	700	5	3147	1.10	2.60
AC-13	KOH	1:6	700	7	2347	0.64	2.73
AC-1@CD	-	-	-	-	1510	0.12	3.68
AC-4@CD	-	-	-	-	135.2	0.09	3.87
AC-9@CD	-	-	-	-	2580	0.80	2.73
AC-12@CD	-	-	-	-	3284	1.20	2.58

**Table 2 materials-16-00689-t002:** Comparison of a few reported materials for adsorption of Cu^2+^.

Material	S_BET_ (m^2^ g^−1^)	Adsorption Capacity (mg g^−1^)	Adsorption Equilibrium Time (min)	Ref.
Biomass carbon residue	-	23.10	120	[25]
AC from rubber tire preparation	-	12.44	120	[26]
AC from Malawian baobab shell	1089	3.083	60	[27]
AC from pistachio	-	277.8	180	[28]
AC-12	3147	49.35	35	This work
AC-12@CD	3284	36.55	30	This work

**Table 3 materials-16-00689-t003:** Fitting parameters of Langmuir and Freundlich isothermal adsorption models.

Material	Langmuir Model			Freundlich Model		
	K_L_ (min^−1^)	Q_m_ (mg g^−1^)	R_L_	K_F_ (L g^−1^)	n	R_F_
AC-12	0.036	61.43	0.9758	8.74	2.4	0.9352
AC-12@CD	0.050	60.35	0.9791	6.13	2.1	0.9656

**Table 4 materials-16-00689-t004:** Fitting parameters of adsorption kinetic model.

Material	Quasi-First-Order Kinetic Adsorption Curve Model		Quasi-Second-Order Kinetic Adsorption Curve Model	
	K_1_	R_1_	K_2_	R_2_
AC-12	0.013	0.9633	0.016	0.9998
AC-12@CD	0.010	0.9593	0.004	0.9961

**Table 5 materials-16-00689-t005:** Comparison of a few reported materials for fluorescence detection of Cu^2+^.

Material	Detection Range (μmol L^−1^)	LOD (μmol L^−1^)	Ref.
CD from Peanut shells	0–50	4.80	[36]
CD from Sago waste	0–48	7.80	[37]
Amino hydroxyl-doped CD	20–80	5.00	[38]
CD from triolein	0.5–10	0.20	[39]
AC-12@CD	10–50 × 10^−3^	1.69 × 10^−4^	This work

## Data Availability

Data available on request from the authors.

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
