# Peer review of "Preparation of Ganoderma Lucidum Bran-Based Biological Activated Carbon for Dual-Functional Adsorption and Detection of Copper Ions"

_materials, 2023, doi:10.3390/ma16020689_

Round 1

Author Response

Reviewer #1:

  1. In section 2.2.1. Preparation of AC, the authors write "The resulting product was dried to a constant weight at 60 ℃ and named as AC@CD." How was the mass of the sample measured? How was the drying stage carried out?

Response: Thank you very much for your valuable comments. Drying means that the weight difference of the sample after two consecutive drying or burning was less than 0.3 mg, and the second and subsequent weighing after drying to constant weight shall be carried out after drying for 1 h at 60 ℃. And based on your comments, we replaced lines 129-130 with " Then, 0.01 g of AC reacted with 0.01 g of CD (after freeze-drying for 24 h) at 25 ℃ for 24 h. The resulting product was dried to a constant weight for 24 h at 60 ℃ and named as AC@CD". (Page 5, lines 129-131)

  1. To obtain a CD, the authors use thiourea. What is the role of sulfur in this process?

Response: The fluorescence intensity of carbon dots was affected by the surface groups, and sulfur doped carbon dots prepared by thiourea have better fluorescence performance.

  1. The section on adsorption is titled: "2.4. Adsorption of metal ions by AC@CD". However, the text provides methods and calculations of adsorbed characteristics only for AC.

Response: Thank you very much for your valuable comments. Based on your comments, we replaced Line 147 with " Adsorption of metal ions by AC". (Page 5, line 147)

  1. The authors report in detail about the influence of the activator nature on the change in the specific surface area of AC, but do not explain why when using an alkaline surface area and porosity increases by orders of magnitude, but not with an acid activator?

Response: Thank you very much for your valuable comments. Based on your comments, we changed with " Potassium hydroxide could inhibit the formation of tar, thus achieving good activation effect at low reaction temperature." (Page 8, lines 226-228)

  1. How does the activation temperature (500, 700 and 900 °C) affect the morphology of AC? Any images of SEM?

Response: The activation temperature (500, 700 and 900 °C) had little effect on the morphology of AC, because the specific surface area of AC prepared at these temperatures is similar, so SEM was not taken. Thanks!

  1. The authors should provide pictures of the SEM with a large magnification, on which honeycomb-like porous structures will be visible. In addition, there are not enough GB images to show the preservation of the fibrous structure after carbonation and activation.

Response: Thank you very much for your valuable comments. Based on your comments, we characterized the sample, and changed the manuscript again. (Page 10, lines 255, 259)

  1. When discussing the adsorption capacity of AC and AC@CD composites, a direct

dependence of the adsorption capacity of the samples on their specific surface area is taken into account. For ease of the material perception and clarity, in table 2 it is necessary to present the values of SBET.

Response: Thank you very much for your valuable comments. Based on your comments, we changed with "Although the large specific surface area usually means that the material has good adsorption capacity, the adsorption capacity of a single material is not only related to the specific surface area. The surface functional groups also affect the adsorption capacity of AC." (Page 13, lines 324)

  1. It is necessary to cite and briefly present some recent work on the production of porous carbon materials from biomass eg. Journal of Energy Storage. 2022. V.50. 104225:1-28. DOI: 10.1016/j.est.2022.104225; Fuel Processing Technology. 2022. V.226. 107076:1-11. DOI: 10.1016/j.fuproc.2021.10707

Response: Thank you very much for your valuable comments. Based on your comments, we quoted in appropriate places. (Page 3, lines 84 and 86)

Reviewer 2 Report

Manuscript ID: materials-2111811

Baoying Wang and co-authorsreported "Preparation of Ganodermalucidum bran-based biological activated carbon for dual-functional adsorption and detection of copper ions". Although the topic is interesting, but some important aspects were not performed. Following comments should be addressed before possible consideration for publication in worthy Journal of Materials. I believe it will not take a long for the authors to work on this revision. My comments are,

1.     In abstract various abbreviations were used. Use full form when write 1st time and then abbreviation be used throughout the manuscript.

2.     In introduction more literature should be reviewed and some latest adsorbents should be discussed here to enhance the novelty of work like, Inorganic Chemistry Communications 145 (2022) 110008, Surfaces and Interfaces 34 (2022) 102324.

3.     Why authors used two types of labels for equations like A1, A2…….. and (1), (2), ……

4.     XRD peaks in Fig. 4b are not clear. provide more clear picture.

5.     Effect of contact time and adsorbent dose should be analyzed

6.     Proposed mechanism of adsorption of Cu2+ onto AC-12 and AC-12@CD should be discussed in separate sub heading.

7.     Reusability test should be performed

8.     FTIR analysis after adsorption should be carried out to check the structural changes of AC-12 and AC-12@CD.

9.     There are so many typo grammatical errors in whole manuscript,should be revised by some native speaker and formatting should be checked.

Author Response

Reviewer #2:

  1. In abstract various abbreviations were used. Use full form when write time and then abbreviation be used throughout the manuscript.

Response: Thank you very much for your valuable comments. Based on your comments, we had changed. (Page 2, lines 32-44)

  1. In introduction more literature should be reviewed and some latest adsorbents should be discussed here to enhance the novelty of work like, Inorganic Chemistry Communications 145 (2022) 110008, Surfaces and Interfaces 34 (2022) 102324.

Response: Thank you very much for your valuable comments. Based on your comments, we had quoted in appropriate places. (Page 13, lines 334, 336)

  1. Why authors used two types of labels for equations like A1, A2…….. and (1), (2), ……

Response: Thank you very much for your valuable comments. Based on your comments, we had changed one type.

  1. XRD peaks in Fig. 4b are not clear. provide more clear picture.

Response: Thank you very much for your valuable comments. Based on your comments, we had changed. (Page 11, lines 279)

  1. Effect of contact time and adsorbent dose should be analyzed

Response: According to your comments, we have changed with "And that of AC-12@CD increased within 30 min, so the equilibrium time was 30 min. The adsorption time was shortened due to the increased adsorption sites of the modified materials. With the increase of time, rate of pollutants removed by adsorbent increases continuously, but the growth rate gradually slows down until the adsorption equilibrium is reached. In the early stage of adsorption, the adsorption is faster because the material mainly complexation reaction with metal ions." (Page 12, lines 307-312)

  1. Proposed mechanism of adsorption of Cu2+ onto AC-12 and AC-12@CD should be discussed in separate sub heading.

Response: Based on your comments, we have changed. (Page 12, lines 297. Page 13, lines 325. Page 14, lines 338)

  1. Reusability test should be performed

Response: We have conducted repeated experiments in our previous work, but the results are not good, which was also a problem for further exploration in the next experiment. Tanks!

  1. FTIR analysis after adsorption should be carried out to check the structural changes of AC-12 and AC-12@CD.

Response: Thank you very much for your valuable comments. Based on your comments, we had changed. (Page 10, lines 267-268. Page 11, lines 280)

  1. There are so many type grammatical errors in whole manuscript, should be revised by some native speaker and formatting should be checked.

Response: Based on your comments, we had changed. Thanks!

Round 2

Reviewer 1 Report

The paper has been improved and can be published in the journal Materials. The authors need to check the correct numbering of the paper sections.

Reviewer 2 Report

Accept